Efficient data transmission on wireless communication through a privacy-enhanced blockchain process

http://orcid.org/0000-0001-8508-6066 Aluvalu Rajanikanth 1 rajanikanth.aluvalu@gmail.com
Kumaran V. N. Senthil 2
Thirumalaisamy Manikandan 3
Basheer Shajahan 4
Ali aldhahri Eman 5
Selvarajan Shitharth 6
1 Department of IT, Chaitanya Bharathi Institute of Technology , Hyderabad , India
2 Department of ECE, SRM Institute of Science and Technology , Tiruchirappalli , India
3 SIMATS Saveetha School of Engineering, Saveetha University , Sriperumbudur , India
4 School of Computing Science and Engineering, Galgotias University , U.P , India
5 Computer Science and Artificial Intelligent Department, Collage of Computer Science and Engineering, University of Jeddah , Jeddah , Saudi Arabia
6 Department of Computer Science, Kebri Dehar University , Kebri Dehar , Ethiopia
Yu Lisu
Electronic publication date: 2023 Apr 21
Publication date: 2023
Volume: 9
Electronic Location ID: e1308
Received 2022 Aug 11; Accepted 2023 Mar 2
Copyright: © 2023 Aluvalu et al.
Copyright year: 2023
Copyright holder: Aluvalu et al.
License: This is an open access article distributed under the terms of the Creative Commons Attribution License, which permits unrestricted use, distribution, reproduction and adaptation in any medium and for any purpose provided that it is properly attributed. For attribution, the original author(s), title, publication source (PeerJ Computer Science) and either DOI or URL of the article must be cited.
License URL: https://creativecommons.org/licenses/by/4.0/

Keywords: Blockchain, Data management, Gradient boosting, Hybrid microwave transmission, Wireless technology

Funding: The authors received no funding for this work.

==============================
In the medical era, wearables often manage and find the specific data points to check important data like resting heart rate, ECG voltage, SPO2, sleep patterns like length, interruptions, and intensity, and physical activity like kind, duration, and levels. These digital biomarkers are created mainly through passive data collection from various sensors. The critical issues with this method are time and sensitivity. We reviewed the newest wireless communication trends employed in hospitals using wearable technology and privacy and Block chain to solve this problem. Based on sensors, this wireless technology controls the data gathered from numerous locations. In this study, the wearable sensor contains data from the various departments of the system. The gradient boosting method and the hybrid microwave transmission method have been proposed to find the location and convince people. The patient health decision has been submitted to hybrid microwave transmission using gradient boosting. This will help to trace the mobile phones using the calls from the threatening person, and the data is gathered from the database while tracing. From this concern, the data analysis process is based on decision-making. They adapted the data encountered by the detailed data in the statistical modeling of the system to produce exploratory data analysis for satisfying the data from the database. Complete data is classified with a 97% outcome by removing unwanted data and making it a 98% successful data classification.

Research introduction

Wireless communication has been vital in data transmission using the network-based Blockchain and security purposes. The application and the remote act as the appropriate rapid deployment of the data transmission proceeded. The 6G mobile system’s remote access also manages the advanced enhancement of the wiles communication (Alhalabi et al., 2023). This makes the advantage of the features and the application of IoT technologies. Then the data is monitored using the data devices (Manoharan et al., 2022).

Also, the gradient boosting method, the type of diseases, and the curing technique can be analyzed by the data stored in the big data using the gradient boosting process. Only the established person can access the patient’s details and treatments (Peng et al., 2021). The authentication process utilizing blockchain technology is discussed, especially in “Proposed authentication process using blockchain technology.” This is the actual function that blockchain technology plays in effective wireless data transmission. A decision regarding the growth of mobile and wearable devices is evaluated decision-based on wireless communication. This evaluation manages the recent trends in wireless communications. This communication contains the factor to access the process of security among the data. This application permanently stores the data as a copy of the database. All people can access the data for phone calls (Dubé & Wen, 2022).

Research contribution

This evaluation tracks the personal details of the confidential data stored in the database. The contribution of the article is as follows based on the wireless communication a) The monitoring of the data transmitted from the wearable devices is analyzed, and the decision-making is done using gradient boosting.

b) The location and the threatening person can be found using the millimeter wave. This manages the location and details tracking.

Research chapters discussion

The introduction discussion presented in “Research Introduction” summarises the research’s basic methodology in the introduction style, and “Literature Survey” conducts a literature evaluation of the previously published article. The suggested system has been used to develop the procedures in “IoT Founded Health Monitoring of Patients” further. The results are presented in “Results Examining Resemblances of Performance”, and the study’s conclusion is provided with suggestions for further research in “Conclusion”.

LITERATURE SURVEY

Ometov et al. (2021) study discussed wearable technology and proposed challenges facing technology development. The process of managing the state-of-the-art has been proposed. Also, implementing this in the proposed system, the current and historical data have been analyzed for better decision-making. Cheng et al. (2021) manuscript study discussed the human-computer interaction and proposed the principle of monitoring the environmental factor. The transmission application manages the wireless technology for demanding the data, which is elaborated in the wireless communication for the diverse antennas and the beamforming of the data. By using this method in the currently proposed system, the system’s formation to make wireless communication and decision-making has been elaborated by Wu et al. (2022). Gao et al. (2021) study discussed the communication-based factor for accepting the transmitting control effects to manage the anomalies which affect the abnormal physical aspect to make the decision based on the intrusion detection method. An effective machine learning algorithm has been proposed to enable this in the currently proposed system (Tran et al., 2021). Juneja, Pratap & Sharma (2021) study discussed the 5G technologies that manage the design and the frequencies for the state of work in the semiconductor for analyzing the national communication network to analyze the system.

The cellular system tracking and the formation of the technology for the CMOS technology. Using this millimeter frequency, the frequency of mobile phones has been proposed to find the crime person. Dilli (2021) study the multi-user performance analysis using the MIMO hybrid beamforming system for analyzing the frequency band based on the procedures for the frequency analysis. Also, by using this in the currently proposed system, the generation between the wave frequencies and the multi-user propagation factor to produce the frequency of the tele calling can be presented. Makimoto & Yamashita (2001) study discussed the frequency of traveling the data transmission for the data transmission and the communication between the wireless and the cloud data. This analyses the frequency of the system to analyze the crime data. By implementing this in the currently proposed method, the data is analyzed using the millimeter wave is done. Hannachi & Slimani (2022) In this study, the microwave allocation and the RF for finding the frequency to detect the electronic devices have been unable to find the crime person. Also, the threat of phone frequency has been found. Also, by implementing this in the currently proposed system, the frequency has been found concerning the location for traction, the exact person, and the location for the analysis. Prabakaran & Ramachandran (2022) discussed that the blockchain-based, cloud-based safe method of conducting financial transactions had been made available here.

Sajja et al. (2021) discussed that blockchain-based agricultural and sector information allows clients to have faith in their data security while enabling global employment and heterogeneity. The data is related to the pharmacokinetics needed to collect the patient’s data and the worry about managing the privacy of the data, which is identifiable of the data that exposes the implementation of the blockchain-based data can be managed to the data that is accessed for the adoption of the system to produce the informatics solution and the stakeholders to produce the privacy worry and the storage movement that enables the genomic and the contracts which are proposed by Albalwy et al. (2022). Moreover, Alssaheli et al. (2022) research distributes the nonprogrammable settings and incoming load balancing, enabling the existing load balancing and cloud facilities, which control traffic congestion. Mobile device data can influence load balancing, and incoming data is examined depending on the internet and blockchain security protocols.

Iot founded health monitoring of patients

The Internet of Things (IoT) represents a rapidly growing network technology of connected objects which has the potential to collect and share real-time data using embedded sensors. This process is designed as a compact wearable device that helps collect real-time health data like blood pressure, heart rate, and so on to identify the disease. It collects, filters, and archives long-term activity data and physiological data using various sensors. In healthcare sector, a type of swarm intelligent algorithm is implemented in the recognition procedure with a modified fitness function and it is termed fruit fly optimization (FFO) (Shitharth et al., 2023). The disease can be identified by evaluating the extracted data and its simulations before it gets grievous (Selvarajan et al., 2022). By providing good medical care, patients heal faster as soon as possible in the beginning stage. As several devices generate a considerable amount of data, it is not easy to store and manage, so cloud technology is adopted. These collected data are analyzed to find records simulations to predict the patient’s conditions or find a particular disease based on symptoms. After analysis, a gradient boosting method is used to decide the disease (Lee et al., 2021). As the health pieces of information are highly confidential, it has been kept secret to prevent data theft by providing access only to an authorized person; a doctor or consultant patient can access the data for monitoring and decision-making.

In Fig. 1, physiological and activity data are collected through embedded medical sensors to form a wearable device. All these extracted data are transmitted through wireless transmission to store the information in the hospital’s cloud server; these signals communicate with the database. The controller processes signals from patients to find the location of particular patients and stores data in an exact location (Liu, Wang & Xiao, 2021). Doctors or consulters supervise these data to avoid risk; thus, in case of emergencies, the controller generates an indication message in the form of an alarm that helps doctors provide timely treatment.

Figure 1 Wearable sensor for motoring patient’s health monitoring system.

Gradient boosting method for decision making

The gradient boosting (GB) is a machine-learning method. This method is used to predict disease or syndrome based on the acquired data through sensors with the help of IoT. This method collected the vulnerable predictive trees and combined them with a machine until a prediction became closer to a ground truth value (Cao, Zhang & Cao, 2021). This recursively splits the tree, and so it is also known as recursive portioning (Metwally et al., 2021).

Ensemble learning is a model used for predicting with more models, such as classifiers or experts, which are strategically generated and combined to solve a particular computational intelligence problem. In the boosting method, random data samples are considered, fitted with a decision tree model, and then trained each model sequentially (Ometov et al., 2021). It improves the mistakes made in the previous learning process through the next learner. A weak learner is used to perform slightly better than a random chance. The development of new weaker learners for handling the rest of the problematic comments is developed from stress (Kumar et al., 2021). Gradient boosting improvised upon some of the features of AdaBoost to create a more robust and efficient algorithm. Expanding is used to lower errors in medical data predictions, such as predicting disease risk using the collected data through wearable devices, as illustrated in Fig. 2.

Figure 2 Proposed enhanced gradient boosting method for decision making.

Proposed enhanced gradient boosting decision

In the gradient boosting decision tree method, a combination of many weak learners extracts one stronger learner. The weak learners fit in such a way that each new learner does into the residuals of the previous step as the model improves (Padmaja et al., 2021). The final model or tree represents the result of each stage, and the most robust learner is obtained.

Gradient descent algorithm that assures the occurrence of gradient descent boosting tree. A gradient-boosting decision tree combines multiple weak classifiers to bring one strong learner. The vulnerable learners are referred to as individual decision trees. Due to this, the boosting algorithms give an accurate output and take more time to learn (Lv et al., 2021). The patient’s health is monitored using wearable devices acquired approximately every second. For each validation process, the inputs are divided into 80% training data and 20% testing data.

Algorithm for making decisions on the collected data

For each input	
Initialize; Fa0	
Fa0 = Argmin∑k=1K⁡L(bk,∝)	
For n ∈ [1−N] & k ∈ [1−K]	
Compute residuals	
Rnk=− [∂L(bk,F(ak))∂(ak)]Fa=Fan−1	
End for	
Create rin node	
For i=1…In	
Compute βin= Argmin∑k=1K⁡(b∗k−∝)	
Compute ∝in= Argmin∑k=1K⁡L(bk,Fakn−1+∝)	
Update Fa0	
Fa0= Fakn−1+V∑i=1In∝inl(aϵrin)	
End for	
End for	

Let take { xk,yk} where k ∈ [1−K] is an input dataset. Set the primary learner as h(ak) ∈ ak = { a1k, a2k … apk} Where the P denotes the count of predicted variables. yk Represents the output as a predicted label. The procedure for gradient-boosting decision trees is as follows. Initially, address the constant value ( ∝) of the model as,

(1) Fa0=Argmin∑k=1K⁡L(bk,∝)

The number of n iterations belongs to [1−N]. Then computes the gradient direction of residuals as follows,

(2) Rnk=−[∂L(bk,F(ak))∂(ak)]Fa=Fan−1

The essential learner h is utilized to fit sample data and get the initial model. According to the least square approach, the parameter βin of the model is obtained as,

(3) βin=Argmin∑k=1K⁡(b∗k−∝)

The loss function is minimized. According to Eq. (4), a new step size of the model, namely the current model weight, is calculated.

(4) ∝in=∑k=1K⁡L(bk,Fakn−1+∝)

The updated model is as follows:

(5) Fa0=Fakn−1+V∑i=1In∝inl(a∈rin)

When the raw data has limited dimension and size as input, the feature branch points information must be computed multiple times. Using an excellent learning rate will make optimization easier (Sprangers, Schelter & de Rijke, 2021). Still, it is usually supposed to be a smaller value, such as 0.1. This shows how the framework of gradient boost predicts complex targets, such as finding the disease or syndrome of the patient with the help of the sensors with IoT (Velthoen et al., 2021).

Blockchain for healthcare

As the digitalization process is disseminated speedaily, the attacks of cyber security and data breaches have increased; thus, information about patients becomes more complicated than before, so the highest standard of privacy for data is more critical in the healthcare system. In case security is breached, the individuals whose health information was inappropriately accessed face several potential harms (Fu et al., 2023). Thus, to increase the system’s safety, the authentication of data using blockchain technology is incorporated into this system. Some features for authentication using blockchain, namely the decryption keys and signing keys on the device, encryption and verification keys, are stored on a blockchain, and it protects data against cyber-attacks (Le et al., 2021).

For an authorised individual to participate in the review process, the user’s identity is necessary. Usually, the authentication is based on the user id and password. Still, blockchain uses an infrastructure of public keys and smart contracts as authentication processes based on certificates, as illustrated in Fig. 3. This removes human factors from the authentication process. These certificates were formulated using a key size higher than a size of a password, as 2,048 bits. These certificates contain the expiry date to reduce the risk of long-drawn-out exposure. Blockchain technology can leverage smart contracts to validate user certificates and effectively mitigate the risk of a single point of failure (Zou et al., 2021).

Figure 3 Proposed authentication process using blockchain technology.

Thus, to restrict access to a particular function, a method of authorization is considered to mitigate the risk of information disclosure for unauthorized users. The restriction of help to users is based on the distributed list of points: action, object, contextual, and subject. The action attribute consists of activities that allow users to perform, like retrieving data and updating records. The object attribute is used to implement which type of object can be accessed by a user or doctor that is medical data related to a particular department (Diao, Wang & Gong, 2022).

At last, the subject attributes denote a user’s request for data access. These attribute types are encoded in a smart contract and shared across the blockchain network. For recording each action, a user identity is used on the network of the blockchain method. A transaction is used to add, update, or retrieve data from a database. This is also used for sharing information between authorized users. An administrator or doctor creates the flow of information in the authentication-based blockchain method and system-based health records and revokes digital certificates (Zhou et al., 2022). A CRL gets generated while revoking a user’s digital certificates. The peers in the blockchain network receive a CRL update command to update the CRL stored on the peers. The users who require access can authenticate with healthcare records by giving certificates to the system.

Figure 4 shows the centralized approval architecture with distributed blockchain method for revokes digital certificates from the distributed blockchain method. These certificates are then shared with users and received by peers in the network; they are established by running a smart contract that validates the credentials and returns a response. When the above-stated conditions are valid, the system allows access to the user; otherwise, it denies access to users (Aljuhani et al., 2022). The privacy preservation of hospital data uses blockchain technology that secures and maintains the hospital data as a ledger. This is referred to as a distributed database; a group of individual people updates that. In this technology, each user or device is named a node. The user is a patient, the wearable devices send the data to the peer network such as the hospital node, and these electronic health records of a patient data are immutable. When updating the record initially, it needs a user for the process.

Figure 4 Centralized approval architecture with distributed blockchain method.

Data structures created by blockchain technology include built-in security features. It eliminates duplication, streamlines procedures, eases the load of audits, boosts security, and guarantees data integrity. It keeps a secure, decentralised log of cryptocurrency transactions. It has a distinctive and secure digital identification reference that enables the distribution and recording of digital information without its editing.

From above figure a new distributed blockchain aspect diagram concerning the healthcare system and respective public keys smart contracts with 2,048 bits for revokes digital certificates. It will maintain the adequate security verification of the hospital data using privacy preservation of hospital data based on blockchain technology. This will be illustrated in the following and algorithms.

Pseudocode for the security verification of the hospital data using privacy preservation of hospital data based on blockchain

Initially, the sensor collects data from a patient	
Data. Store→EHR	
EHR. Store→blockchain ()	
For each user	
Create digital certificate	
revokes. Certificate→(CA) certificate authority	
generate URL	
end for	
for each user authentication	
validate certificate	
user→peer authenticates	
if (CA sign present in the certificate) && (user present in CRL)	
allow user access	
else	
refuse user access	
end if	
end for	
Output: allow or refuse the user access	

The permissioned blockchain technology needs authentication access to verify certificates and intelligent contracts. Systems can identify user access and restrict unauthorized access to system functions (Abyaneh, Zorba & Hamdaoui, 2022). Generally, this research system uses password authentication, but this research system is replacing this authentication with certificate-based authentication. This certificate-based authentication removes human interference during the process of authentication. These certificates are created with a critical size of 2,048-bit, which is larger than the size of an average password and includes the expiration date, reducing the risk of prolonged data exposure (Beniiche, Rostami & Maier, 2022). Blockchain technology has the advantage of smart contracts that validate the user certificates and can efficaciously mitigate the risk of a single point of failure.

The administration creates digital certificates to maintain the electronic health record (EHR) system, and the authority revokes issues in the certificates. The certificate revocation list (CRL) is then generated among all network peers (Park et al., 2023). Before that, the system needs to authenticate by digital certificates (Khan et al., 2022). The certificate is passed to the client, which is then sent to a peer in the network. The peer authenticates the user by running the smart contract, which contains information about the user/doctors, action attribute, object attributes, and contextual and subject attributes (Xu et al., 2021). That validates the certificate and returns a response. This response influences the authentication condition of the user’s access as allowing or refusing.

Algorithm for improving the data transmission method for wireless transmission

Input: Triangles (G), L-link, R-result	
Output: Predict the path	
Require: G = (V, E)	
Load(G)	
Triangles = Find Triangles (G)	
return null	
initialize: N = size (Triangles), k= 1	
While (k ≤ N)	
New_L= Predict (Triangles) [i]	
For each_L ∈ New_L	
If (R_contains L)	
R [L]++	
Else	
R = R + L	
End If	
End For	
increment k	
End While	
R = Sort Descending Result	
Return R	

Tracking the location of a threatening person

Medical records can be misused unless coded to hide patients’ identifying information. If a patient has been treated for a particular disease and their medical records are not held in confidence might be insecure. Therefore, these records are secured using a piece of code that is difficult to access by unauthorized people. Although the information is confirmed, there might be a chance to steal medical data with personal information by a trusted person who knows a private key or by a hacker who can break the security code and access the data illegally to make money or for some other reasons by threatening patients (Akter et al., 2022). They protect data whenever information is accessed illegally or when a patient is intimate about any threatening call, message, or another. There is a chance of doctors getting threatened as the details of doctors are also stored in the medical record based on their treatments and so on (Kumar et al., 2022).

When a threat is reported, management tracks a person’s location using hybrid microwave transmission, which combines millimeter-wave and microwave. The millimeter wave is a spectrum band with a wavelength between 10 to 1 mm with a highly high-frequency band. Thus, millimeter waves can hardly be used for long-distance applications. A microwave uses a beam of radio waves in the microwave frequency range to transmit data between two locations which can be located within a few feet or meters or miles or kilometers. It also has higher bandwidth and a limited line of sight mode (Hsiao & Sung, 2022). Both are used in communication, but millimeter wave has a higher frequency range than microwaves (Wan et al., 2022). Thus by combining these wave transmission methods, the distance covered, line of sight, amount of data carried from sender to receiver, and speed of data transmission can be effectively increased; this is called hybrid microwave transmission. In this system, hybrid microwave transmission tracks a person at the time of the call. If a person is not disturbed through the ring, it might be through a message or any other method. Considering it, a person can be tracked faster to a considerable distance (Althubiti et al., 2022). To find the location, this research uses a MapQuest open-source mapping service connected to a tracking system to view the exact location of a threatening person. It provides street details and directions for different countries (Deng et al., 2022). The user is provided access to a map of any place in the world. Through a gas price feature, the user can find information about a particular location because the person who threatens a patient might be in any of the locations, and it also allows one to find a person’s current location.

The link prediction method improves a data transmission’s efficiency (Li et al., 2022). It is used to route the link prediction based on the behavior of a mobile node. During the person’s movement, various nodes might meet neighboring nodes. When the node moves, the path’s weight increases from the source node to the information interaction node. The weight increases are directly proportional to the quality and amount of data obtained. The weight depicts the coordination between nodes in the social process that enables other nodes to find the optimal path to get information faster and to track the transmission target node as quickly as possible to complete the community structure reconfiguration (Yang et al., 2022).

In the social graph, the probability of a mobile node i from the source node j to interaction node k, as shown below path prediction (Pr) equation,

(6) Prijk=WβjmCαjm∑mϵMji⁡(WβjmCαjm)

It indicates where the Wβjm denotes the weight of the path and Cαjm The cost of moving from node j to k corresponds to the coefficient of basic parameters. Mji denotes the critical points between nodes i to j. This model assumes that all edges in the social network graph have initial weights. The optimal solution is that the target node must obtain information (Zhou et al., 2022). The link where people receive information will increase the weight, as shown in the below equations

(7) ΔWijk=1Ciwhere,(j,k∈Li)

(8) ΔWijk=0,otherwise

(9) Wjk=Wjk+∑i=1t⁡ΔWijk

Li denotes the path traversed by the mobile device that from the information source, the length represented as Ci. Where t is the total count of mobile nodes, which indicates the search for the shortest path from the starting point to the information source. Assuming that each edge’s weight will gradually decrease, the more the path is traversed, the greater the weight.

Then analyze the reduction mechanism of path weights, as shown in the below equation.

(10) Wjk=(1−p)Wjk

The Eq. (1), the mobile devices are moving from node j to node k from the equation

(11) Prijk=(1Wjk)β∑mϵMji⁡(1Wjk)β

where wjk is the weight of the corresponding path, and the coefficient β is the primary parameter of the model, which is set to 1. Comparing Eq. (6) can see that α = 0, which is inversely proportional to the weight value. If the current memory module is filled, it will be covered one by one according to the order of the arrow’s direction on the way.

Additionally, the accuracy of each node is confirmed each time this research acquires three additional nodes. Let us say that one of the memory modules already has the written data. It is assumed that a triangle relationship has been discovered in that situation (Dhieb et al., 2020). All pathways start with a weight of one. The weight of each direction is raised by one each time a new triangular connection is discovered. The edge’s weight is increased by one if the present pathway is a part of an existing triangle connection. Each path’s importance will not diminish with time.

The patient consults a doctor for any biological changes found in the body or due to any disease symptoms. Then, the doctors gather all the information from patients through a lab test, MRI, sensor data, and so on, providing treatment accordingly. All these details, like test reports, sensor data, scan reports, personal information, treatment information, etc., are stored in a database as electronic records. These records can be accessed by both the patients and doctors and then managed by hospital management. When a patient receives a threatening call or message, they intimate control and track the location of a person through Hybrid microwave transmission for location tracking and MapQuest for providing an exact location in map format to the management, as illustrated in Fig. 4.

Results examining resemblances of performance

The IoT devices continuously sense and collect data on a user’s health that helps the user’s health status. In this work, the personal health data of a user is collected by wearable devices. Human heart rate, blood pressure, temperature, blood oxygen rate, the glucose level in the blood, and other physiological data activities like motion, running, and exercise data are collected using sensors like heart rate sensors, blood pressure sensors, motion sensors, temperature sensor and similarly, can be used to collect the physiological data of a user. To assess our proposed approach, this research takes samples of 200 users containing each user’s sensor data. For evaluating the efficiency of the predicted health data, this research uses confusion metrics to classify the performance. Also, the following parameters are used to enhance the processing scheme concerning the black chain technology parameters (Table 1).

Table 1 Comparison amount various classification methods with the proposed hybrid gradient boost technique.

Methods	Naive Bayes	Decision tree	Proposed hybrid gradient boost technique	
Number of folds	Precision	Recall	F-measure	Precision	Recall	F-measure	Precision	Recall	F-measure	
1	84.23	89	86.54	85	94	89.27	96.3	97	96.64	
2	83.34	90	86.5	85	94	89.27	99.56	99	99.27	
3	82.64	91	86.61	83.64	95	88.95	99.23	98	98.61	
4	81.09	89	84.86	85	95	89.72	97.5	98	97.74	
5	80	89	84.26	85	95	89.72	98.9	99.2	99.04	
6	84.21	91	87.47	86	96	90.72	98	99	98.49	
7	84.23	91	87.48	87.1	96	91.33	97	99.5	98.23	
8	82.5	89	85.62	87.3	95	90.98	97	98	97.49	
9	84.4	91	87.57	87.9	94	90.84	97.4	98	97.69	
10	84.4	89	86.63	84.4	95	89.38	98.4	99.2	98.79	
11	82.5	91	86.54	84.5	95	89.44	98.5	99.2	98.84	
12	83	91	86.81	88	94	90.90	97	99	97.98	
13	79.1	90	84.19	83	96	89.02	97.5	99.3	98.39	
14	79.2	90	84.25	85	94	89.27	97.6	99	98.29	
15	79.3	91	84.74	83.9	96	89.54	97.9	98	97.94	
16	80.2	91	85.25	83.89	95	89.10	96.62	98	97.30	
17	82.3	90	85.97	84.6	96	89.94	99.3	99.3	99.3	
18	80.2	89	84.37	87.62	95	91.16	97.2	98	97.59	
19	81.7	91	86.09	87.56	94	90.66	97.7	99.2	98.44	
20	83.6	90	86.68	84.3	94	88.88	99.4	99.4	99.4	
Note:

The gradient boost decision-making classification method is applied to the healthcare data of a user. The trained classifier verifies the dataset and is classified into k-an exclusive mutual unit using the k-fold cross-validation method. K-fold cross-validation is applied with values of 1, 5, 10, 15, and 20 for k.

The confusion metrics have five factors, including four parameters: accuracy, f-score, precision, recall, and execution time of the sensor data. Whenever a researcher concludes something is real when it is actually wrong, this is known as a type 1 error (also called a false positive type I error). A “false alarm” is a false positive. Saying something is untrue when it is truly true, is known as a false negative (also called a type 1 error). four parameters are true positive, false negative, false positive, true negative, because uts useful detect following. Accuracy, Sensitivity and Specificity. The parameters are true positive (tp), which defines the samples data that are correctly classified as abnormal cases, and false negative (fn), which indicates the anomalous samples are classified wrongly as usual. False positive (FP) defines the standard examples that are classified incorrectly as abnormal cases, and true negative (TN) shows the traditional models that are correctly classified. The GB Gradient boosting is a machine learning method. The word boosting represents a method where weak predictions are combined for a more robust prediction method. In this method, the vulnerable predictive trees were collected and combined with a machine until a prediction became closer to a ground truth value. The decision trees are generated through iterative questions to the separation of data and to obtain a decision. This method is used to predict disease or syndrome based on the acquired data through sensors with the help of IoT.

(12) Accuracy=(tp+tn)(tp+tn+fp+fn)

(13) Precision=(tp)(tp+fp)

(14) Recall=(tp)(tp+fn)

(15) F-Measure=2∗(Recall∗Precision)(Recall+Precision)

The gradient boost decision-making classification method is applied to the healthcare data of a user. The trained classifier verifies the dataset and is classified into k-an exclusive mutual unit using the k-fold cross-validation method. K-fold cross-validation is applied with values of 1, 5, 10, 15, and 20 for k. Here the k resampling samples called cross-validation are used to assess machine-learning models on a small data sample. The process contains a single parameter, k, that designates how many groups should be created from a given data sample. As a result, the process is frequently referred to as k-fold cross-validation. It aids in preventing overfitting. We know that the highest performance accuracy is achieved when a model is trained utilizing all the data in a single brief run. We can design a generalized model by avoiding this k-fold cross-validation.

There are typically three methods to choose k: let k equal 25 or 250. It has been discovered via experimenting that choosing k to be 25 or 250 produces good enough results. Let k equal n, where n is the dataset’s size. Thus, the test data set will include each sample. Another option is to pick k so that each split data sample has a sufficient size to ensure statistical representation in the larger dataset. A technique for thorough cross-validation is leave-p-out cross-validation. The training dataset, in this case, contains the p observations (or sample dataset items); everything else is regarded as training data. Figure 5 will make it more straightforward that p is equal to 25, as indicated by the 25 circles in the “test data” section. A unique variation of the leave-p-out exhaustive cross-validation method, when p = 10, is called leave-one-out cross-validation. In the situation of k-fold cross-validation, where k = N, this is likewise a particular case (number of elements in the sample dataset) (Table 2).

Figure 5 Proposed process of location tracking of a threatening person.

Table 2 Comparison performance of various decision-making methods

with the proposed gradient boost decision-making method corresponding to the execution accuracy with execution time.

Decision-making method	Execution time (ms)	Execution
accuracy (%)	Precision	Recall	F-measure	
Gradient_Boost	12.77	97.9	92.92	88.73	98.27	
Naïve_Bayes	72.86	83.72	82.107	90.15	85.92	
Decision_tree	52.35	87.91	85.43	90.2	89.90	

Figure 5 illustrates the various classification methods’ precision values with different k-cross folds. The curves in the graph show that the gradient boost method has the highest precision value of 99% compared to other classification methods. The decision tree and naïve Bayes method have 87% and 84% precision, respectively.

As demonstrated in Fig. 6, the proposed hybrid gradient boost technique method has reached the highest recall value of 99% compared to the decision-making with 92% and the NB with 90% for the recall factor.

Figure 6 Analysis of precision.

As illustrated in Fig. 7 exposes that the gradient boost classifier has gotten the highest f-measure value of 96% compared to the decision-making with 88% and the NB with 83% for this f-score factor.

Figure 7 Analysis of recall.

Figure 8 shows how the suggested hybrid gradient boost approach effectively outperformed decision-making and NB in recall factor, each of which had recall values of 97% and 98%, respectively. As shown in Fig. 8, the gradient boost classifier obtained the highest accuracy score (96%), followed by the decision-making classifier (81%) and the NB classifier (73%).

Figure 8 Result analysis of F-measure.

Table 3 shows the comparison measure of the various decision-making methods with the proposed gradient boost decision-making method corresponding to the execution accuracy with execution time. From the table, the accuracy of the gradient boost is higher than the other methods. After that two algorithms, the decision tree classifier has higher accuracy and execution time than the naïve Bayes method. The naïve Bayes has the most negligible accuracy value compared with other methods.

Table 3 Analysis of average response time data submission and validation.

The results below demonstrate the system’s ability to quickly validate and transmit large volumes of data.

Methods	Decision tree	Naïve Bayes	Gradient boost	
	Data submission	Data validation	Data submission	Data validation	Data submission	Data validation	
No. of data records	Average response time (ms)	Average response time (ms)	Average response time (ms)	Average response time (ms)	Average response time (ms)	Average response time (ms)	
10	20	0.300	11.031	0.173	8.969	0.127	
100	100	4.300	69.040	3.315	30.960	0.985	
1,000	800	8.200	610.875	6.912	189.125	1.288	
2,000	1,000	14.300	808.795	12.591	191.205	1.709	
4,000	1,500	19.000	1,213.193	17.167	286.807	1.833	
6,000	2,500	21.200	2,098.716	19.486	401.284	1.714	
8,000	3,000	24.700	2,585.062	22.981	414.938	1.719	
10,000	3,500	26.800	3,074.727	25.162	425.273	1.638	
20,000	5,500	27.800	4,904.117	26.286	595.883	1.514	
50,000	14,200	29.800	12,811.926	28.337	1,388.074	1.463	

As shown in Fig. 9, the naïve Bayes and decision tree methods were performed with the highest execution time among the other classifiers. Finally, the gradient boost classifier has performed with a minimum execution time of 12 ms over the same dataset. As shown in Fig. 9, the naïve Bayes and decision-making method has been performed with the lowest execution accuracy among the gradient boost classifier. And finally, the gradient boost classifier has performed the highest accuracy of 97.9% over the same dataset. The user needs privacy for their health data and access by the doctor and denies access from third parties.

Figure 9 Result analysis of accuracy score.

As illustrated in Fig. 10, the number of data and average response time for data submission and the number of data and average response time for data validation, respectively. When the data is submitted to the blockchain ledger, this takes time for the process; this processing time must be less or higher. This is only based on the amount of data. If the data count is the lesser time taken for the process is minor; similarly, if the data is large, then the time taken for the operation or response time might increase. Besides, the validating portion of the blockchain follows this condition. The data submission analysis and validation for the existing decision tree took an average response time of 3,212 ms (on data submission). It took an average response time of 17.640 ms (on data validation). This average response time is 10% higher than the existing approach.

Figure 10 Comparison of various classification methods with execution time.

Conclusion

The blockchain method and proposed hybrid gradient boost that was suggested concentrated on data transfer to compare several new technologies employing wireless communication. The use of decision trees and blockchain-based decision-making were used to optimize wireless communication. The recommended system produces the following results in terms of prediction accuracy, execution time, execution accuracy, and average response time of data submission and data validation. The suggested hybrid gradient boost methodology offers the best precision value of 99% compared to the prior classification method. The decision tree and the naïve Bayes algorithm have 87 and 84% accuracy rates, respectively. Over the same datasets, the hybrid gradient boost classifier has the fastest execution time, clocking in at 12 ms. Naïve Bayes and the decision tree method have the most prolonged execution durations compared to the other classifiers. However, naïve Bayes and the decision-making strategy had the lowest execution accuracy when using the gradient boost classifier. Performance for the gradient boost classifier, which offers the highest level of precision, is 97.9%. The research on data submission and validation for an existing decision tree found that average reaction times were 3,212 and 17.640 ms, respectively. The average reaction time of this approach is 10% longer than the present method. In the future efficient data transmission, such as existing File Transfer Protocols (FTP) and Secure File Transfer Protocols (SFTP), will be handled concerning machine learning on wireless communication through an enhanced hybrid blockchain process.

Supplemental Information

Supplemental Information 1 Analysis of Data Submission and Data validation response time for Existing Decision Tree.

Click here for additional data file.

Supplemental Information 2 Analysis of Data Submission and Data validation response time for Existing Naïve Bayes Algorithms.

Click here for additional data file.

Supplemental Information 3 Analysis of Data Submission and Data validation response time for Existing Gradient Boost Algorithms.

Click here for additional data file.

Supplemental Information 4 Analysis of average response time data submission and validation.

Click here for additional data file.

Supplemental Information 5 Complete Blockchain Code.

Click here for additional data file.

Additional Information and Declarations

Competing Interests

Author Contributions

Data Availability

Rajanikanth Aluvalu is an Academic Editor for PeerJ.

Rajanikanth Aluvalu analyzed the data, authored or reviewed drafts of the article, and approved the final draft.

Senthil Kumaran V. N. conceived and designed the experiments, performed the experiments, performed the computation work, authored or reviewed drafts of the article, and approved the final draft.

Manikandan Thirumalaisamy conceived and designed the experiments, performed the experiments, performed the computation work, authored or reviewed drafts of the article, and approved the final draft.

Shajahan Basheer conceived and designed the experiments, performed the experiments, performed the computation work, authored or reviewed drafts of the article, and approved the final draft.

Eman Ali aldhahri performed the computation work, prepared figures and/or tables, authored or reviewed drafts of the article, and approved the final draft.

Shitharth Selvarajan analyzed the data, authored or reviewed drafts of the article, and approved the final draft.

The following information was supplied regarding data availability:

The code is available at GitHub and Zenodo: https://github.com/Shitharth90/shitharthpublic.git; Shitharth90. (2022). Shitharth90/shitharthpublic: v1.0.0 (v1.0.0). Zenodo. https://doi.org/10.5281/zenodo.7423963.

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
