# Peer review of "Efficient data transmission on wireless communication through a privacy-enhanced blockchain process"

_PeerJ Computer Science, doi:10.7717/peerj-cs.1308_

## Round 0.1 · original submission · Major Revisions

After reading the reviewers' comments and the editor's reading, the paper needs to be revised. Please revise the paper and resubmit it after careful revisions. Remember to highlight the revision parts in the revised manuscript.

The comments in detail provided by reviewers are given below.

Reviewer 1 ·

Basic reporting

The narrative and structure of the paper are generally clear and reasonable. The use of blockchain-based wireless communication technology for data analysis in wearable devices and the location of threatening people through hybrid microwave technology are well reflected in the paper.

Experimental design

The experimental design is reasonable. The comparative experiments of the hybrid gradient boosting method proposed in the article, the decision tree method and the naive Bayes method, well reflect the advantages of this decision-making method.

Validity of the findings

From the experimental results, the hybrid gradient boosting method has a good performance in the processing time after data submission and the accuracy of data validation, but some results still need to be revised and re-validated.

Additional comments

1、What does the k-fold cross-validation method in lines 415-418 do for data classification? What does the K value stand for?
2、Please re-examine the description of the conclusions on lines 461-467 for Figures 10 through 12.
3、The description of the summary of conclusions in Section 5 is too repetitive with the previous presentation of the findings, please reorganize the language to describe the conclusions.

Reviewer 2 ·

Basic reporting

1. Ca^jm in formula (6), according to the usage in the following, is wrongly expressed.

Experimental design

2. The last part of Chapter 3.3, where people will die when people are not using the system, should not be counted as dead. What's more, whether the energy of the device needs to be considered, the wearable device should not be judged to be dead if it is out of power.
3.The four parameters are true positive, false negative, false positive, true negative, why not two parameters, for example, true positive is false negative.

Validity of the findings

4.It is difficult to understand whether the simulation Figure 8 is wrong.

Reviewer 3 ·

Basic reporting

Please see the detailed comments in "4. Additional comments".

Experimental design

Please see the detailed comments in "4. Additional comments".

Validity of the findings

Please see the detailed comments in "4. Additional comments".

Additional comments

In this paper, the authors proposed a blockchain-based data transmission method for security improvement in wireless communications. Simulation results demonstrated the effectiveness of the proposed method. The reviewer’s comments are given as follows.

1. The Abstract needs to be revised. In the Abstract, the problem they want to solve and the performance gain by using their proposed method should be clearly stated.
2. In Section 1 and Section 2, the authors should discuss the novelty and differences of the paper compared with existing works. Besides, the motivation of using blockchain should be given.
3. In Section 3
-1) More contents about the blockchain implementation procedure should be given. How does blockchain helps to improve the security?
-2) How to achieve the blockchain verification? What consensus algorithms do the authors use in the paper?
4. In Section 4, the detailed information about the scenario and parameter setup should be given such that readers can replicate the simulation results.
5. The literature review of the paper about using blockchain for wireless communications needs to be enhanced such as the following study can be included in the paper to provide readers a better background.
-“Blockchain-based secure spectrum trading for unmanned-aerial-vehicle-assisted cellular networks: An operator’s perspective,” IEEE Internet of Things Journal, 2020.
6. The reference sequence is disordered, such as the first reference is [3].
7. The paper organization needs to be revised. It is suggested the authors put the figures and tables in the main text to facilitate readers to read the paper.
8. It is suggested that the authors ask some experts to proofread the paper and polish the language.

---

## Round 0.2 · Minor Revisions

From the comments from reviewers, the paper can be accepted now. However, there are still some minor comments from the reviewers. The authors need to address the comments in the final submission version.

Reviewer 1 ·

Basic reporting

There is no problem in the paper as a whole, and the various problems raised have been basically solved. The efficient data transmission method based on blockchain proposed in the paper is also well reflected in improving the security of wireless communication.

Experimental design

The experimental design has not changed much from before, and the questions raised have been revised as required.

Validity of the findings

The modification and reformulation of the experimental results verify that the algorithm proposed in this paper is useful for the data transmission of wireless communication system.

Reviewer 2 ·

Basic reporting

Before that, we have got a better solution to the problems raised.
Now, in Chapter 3.2, when introducing blockchain methods, we can introduce blockchain methods in more detail, such as the consensus mechanism used. First, we should highlight the importance of blockchain.
Secondly, perhaps we can add a picture to introduce the blockchain method proposed in the previous question.

Experimental design

no comment

Validity of the findings

no comment

Additional comments

no comment

---

## Round 0.3 · accepted · Accept

The authors have addressed all the comments from reviewers. No more comments.